# Lipid Formulations and Bioconjugation Strategies for Indomethacin Therapeutic Advances

**DOI:** 10.3390/molecules26061576

**Published:** 2021-03-12

**Authors:** Anna Gliszczyńska, Marta Nowaczyk

**Affiliations:** Department of Chemistry, Wrocław University of Environmental and Life Sciences, Norwida 25, 50-375 Wrocław, Poland; mnowaczyk178@gmail.com

**Keywords:** bioconjugates, indomethacin, lipid-based formulations, prodrugs, drug-phospholipid conjugates, liposomes, nanoparticles

## Abstract

Indomethacin (IND) is a drug which after successful clinical trials became available for general prescription in 1965 and from that time is one of the most widely used anti-inflammatory drug with the highest potencies in the in vitro and in vivo models. However, despite its high therapeutic efficacy in relieving the symptoms of certain arthritis and in treating gout or collagen diseases, administration of IND causes a number of adverse effects, such as gastrointestinal ulceration, frequent central nervous system disorders and renal toxicity. These obstacles significantly limit the practical applications of IND and make that 10–20% of patients discontinue its use. Therefore, during the last three decades many attempts have been made to design novel formulations of IND aimed to increase its therapeutic benefits minimizing its adverse effects. In this review we summarize pharmacological information about IND and analyze its new lipid formulations and lipid bioconjugates as well as discuss their efficacy and potential application.

## 1. Introduction

Indomethacin (1-(*p*-chlorobenzoyl)-5-methoxy-2-methylindole-3-acetic acid) (IND) is an indole-acetic acid derivative which belongs to the group of non-steroidal anti-inflammatory drugs (NSAIDs). This compound is one of the most useful antirheumatic drugs and prime drug recommended for the treatment of gout, collagen diseases, ankylosing spondylitis, osteoarthritis, and acute shoulder pains. Indomethacin was discovered in 1963 and two years later has been approved by the Food and Drug Administration (FDA) as a drug with anti-inflammatory, analgesic, and antipyretic properties for use in the United States [1].

The activity of IND and other NSAIDs is based on the inhibition of the synthesis of prostaglandins (PGs), which mediate in the development of inflammatory reactions in the body. Their mechanism of action is based on blocking the synthesis of prostaglandins by influencing the activity of cyclooxygenase (COX), the enzyme, which is responsible for the conversion of arachidonic acid into PGs (Figure 1) [2]. There are two main isoforms of cyclooxygenase–constitutive COX-1 and induced COX-2. The first one occurs in most of the tissues of the body and participates in the conversion of arachidonic acid to prostaglandin E2 and I2 and thromboxane A2. Therefore, it is characterized by cytoprotective activity toward the digestive system. In addition, it has a positive effect on blood flow in the kidneys and regulates the activity of platelets. Cyclooxygenase-2 (COX-2) is induced by inflammatory mediators (endotoxins, IL-1, TNF-α) and expressed as a result of injury or inflammation [3]. COX-2 is responsible for the formation of prostaglandins that induce an increase of vascular permeability, edema, and pain. Under physiological conditions, there is also the constitutive COX-2 induced by sex steroid hormones. Its presence was also confirmed in the area of healing gastrointestinal ulceration and during *Helicobacter pylori*-induced inflammation [4]. In 2002, a protein called COX-3 was also discovered. It is a variant of the form of COX-1 that arises in the pathway of post-transcriptional modification of COX-1 mRNA. It occurs, in the central nervous system (CNS), and its activity is inhibited by paracetamol, metamizole and used in small doses of NSAIDs, such as diclofenac [5]. In terms of the correlation of selectivity of NSAIDs in relation to cyclooxygenase, they are divided into two major categories: non-selective which block both COX-1 and COX-2 and COX-2-selective inhibitors. The anti-inflammatory effect of these compounds is mainly due to inhibition of cyclooxygenase-2 activity, while the inhibition of COX-1 is responsible for the occurrence in gastrointestinal tract (GI) undesirable side-effects [6].

Indomethacin belongs to the group of non-selective non-steroidal anti-inflammatory drugs. Similarly, to acetylsalicylic acid, diclofenac or ketoprofen, indomethacin also acts on the way of additional mechanisms of formation of pain and inflammation demonstrating the ability to inhibit the passage of leukocytes through the vascular walls to the site of inflammation in the tissues [7]. According to the literature data IND reduces the concentration of leukocytes in carrageenin-induced inflammatory exudates by up to 35 per cent [8,9,10], and the reduction of total numbers of leukocytes by cyclooxygenase (prostaglandin synthetase) inhibitors is proportional to the reduction in edema volume [10]. It was also confirmed that indomethacin has a relatively small effect on leukocyte chemotaxix in vivo at doses which have anti-inflammatory effect and inhibit prostaglandin biosynthesis [11].

The relative potency of NSAIDs increases in the following order: aspirin < phenylbutazone < ibuprofen < naproxen < indomethacin both in the in vitro and in vivo models. This indicates the potent anti-inflammatory effect of indomethacin compared to other NSAIDs and partially is responsible for the popularity of the first three drugs [12]. Indomethacin is currently available in the form of orally administrated capsules as well as rectal and intravenous formulations. This drug is rapidly and completely absorbed in the gastrointestinal tract. Its oral absorption depends on the important inter- and intraindividual variations and is decreased and delayed when indomethacin is administrated with food [13]. In general, 2–3 μg/mL of peak plasma concentrations are detected between 30 min and 2 h in a fasting state [13] and 90% of the drug is bound to albumin and extensively bound to tissue [14]. Peak indomethacin concentration in synovial fluid have been shown to occur approximately 1–1.5 h after peak plasma concentration and is about 25% of these in plasma. However, after 4–5 h the concentrations in synovial fluid are equivalent to those in plasma [15].

Due to the high bioavailability of indomethacin this drug does not undergo significant first-pass metabolism. About 60% of the administered dose is excreted in urine by renal tubular secretion predominantly as glucuronides, while the rest in feces after biliary secretion. Indomethacin undergoes extensive biodegradation via *O*-demethylation, *N*-deacylation, or both reactions (Figure 2). Its main metabolites formed in the liver are *O*-desmethyl-indomethacin (DMI), *O*-deschlorobenzoyl-indomethacin (DBI), *O*-desmethyl-*N*-deschlorobenzoyl-indomethacin (DMBI), and their conjugates with glucuronides [16]. All of them are devoid of anti-inflammatory activity. It has been established that the drug disappearance from plasma is biphasic, with a half-life of 1 h during the initial phase and 2.6–11.2 h during the second phase [3].

## 2. Therapeutic Activity of Indomethacin and Its Mechanism of Action

The first information about non-standard usage of NSAIDs have been raised almost fifty years ago. In 1973 Stoll reported in his initial observation that the daily administration of 100 to 150 mg of oral indomethacin greatly relieved the pain of breast cancer bony metastases and caused almost complete disappearance of a group of metastatic chest wall nodules that ranged in size from 0.5 to 1 cm in diameter [17].

After this report anticancer activity of indomethacin and its mechanism of action have been extensively studied. Summarized data are presented below in Table 1. Wadell and coworkers were one of the first who described the efficacy of indomethacin in reduction of desmoid tumor [18,19]. They have suggested that indomethacin initiates a complex series of chemical effects that impair proliferative capacity of tumor cells and at the same time stimulate immunologic responses. Tumor growth inhibition was confirmed in six of seven patients treated with indomethacin. Moreover, extension of the survival of patients with metastatic carcinoma of the stomach was observed.

Since the publication of these results, extensive research on the usage of indomethacin in the treatment of cancers has begun. Considerable amount of evidence from in vitro studies in animal models of colon cancer and human colorectal cancer cells, compared with results from human clinical trials, has shown that indomethacin has antitumor activity against colorectal cancer. Its action is to prevent the formation of Aberrant Crypt Foci (ACF), which are the earliest developing precursors of epithelial neoplasia and predate the development of colon epithelial polyps [20,21]. A significant reduction in the number of aberrant crypt foci in rats treated with indomethacin at 2 mg/kg per day was observed [20]. Moreover, it was confirmed that indomethacin reduces the size and number of tumors already present in the large intestine [22,23]. A similar effect of its action was also reported in studies conducted on head and neck tumors. The use of indomethacin led to regression and stabilization of tumor development [24]. In the case of skin cancer, IND administration also contributed to the reduction of the tumor mass. In a study conducted on a group of nine patients, complete tumor regression was confirmed in three patients within several weeks of indomethacin therapy. For others patients, partial regression of tumors was observed, and in one patient only beneficial effects of the drug administration was noticed.

Eli et al. reported that a low dose of indomethacin 10–20 μM reduces lung carcinoma cells viability in the in vitro study, whereas 2 mg per kg/day administered to mice effectively inhibits the growth of lung cancer and its metastatic nodules by acceleration of apoptosis and inhibition of cell proliferation. Significantly, IND attenuated metastatic growth even when given only after primary tumor amputation, at which time micrometastases are already present in the lungs. All the indomethacin effects were seen at plasma concentrations of approximately 6 μg/mL, which bears direct relevance to the therapeutic blood levels of this drug in man [25]. Although the Eli results support the notion that COX inhibition is a major determinant in the antitumorigenic effect of IND the molecular mechanism of how indomethacin is involved in inhibiting cancer cell focal adhesion and migration still remines unclear. Brunelli et al. have indicated on the molecular mechanism underlying the pleiotropic activity [26] whereas Guo and coworkers have identified a new mechanism of action for indomethacin–inhibition of calcium influx that is a key determinant of cancer cell migration [27]. It has also been reported that indomethacin can effectively restore TRIAL sensitivity in melanoma cells and act by upregulation of death receptor 5 (DR5) and/or downregulation of survivin [28]. In the last few years, intensive research has also been conducted on the possibilities of using indomethacin in the treatment of castration-resistant prostate cancer [29] and its formulations in hormone-refractory breast cancer [30,31].

In addition, some studies indicate that indomethacin could find application also in the treatment of Alzheimer’s disease (AD). Research in this direction has begun in the 1990s. Clinical trials carried out by Rogers’s team proved that IND administered orally in a dose 100–150 mg/day protects patients with mild and moderate disease against cognitive deterioration. Unfortunately, it was also noted that in the studied group, 20% of patients suffered from side effects of the drug which occurred in the gastrointestinal tract [32]. Other drugs from the NSAIDs group were also tested in this respect, but despite their alleviation of the symptoms of the disease, they were eliminated from further studies due to their strong gastrotoxicity [33].

## 3. Side Effects of Indomethacin and Other NSAIDs

Indomethacin as well as other NSAIDs is characterized by a strong action and a short half-life in the body, which results in its high therapeutic efficacy, but at the same time is responsible for increase of toxicity toward the digestive system and kidneys. It has been estimated that 30–60% of patients who has taken therapeutic doses of IND unfortunately experienced adverse effects and in consequence 10–20% of them discontinued the therapy. The level of possible adverse effects after treatment of IND is dose related [34].

Drugs of this group differ in potency to induce the adverse effects, but several common biochemical actions characteristic for NSAIDs, that contribute to cell damage can be distinguished [35]. NSAIDs as the weak acids, soluble in lipids act locally and interact with the hydrophobic surface of the gastrointestinal mucosa, which leads to the damage of this barrier [36]. This group of drugs inhibit the secretion of prostaglandins, which are not only the inflammatory mediators, but also play a number of physiological functions in the human body. In the intestines, they play a protective role: they stimulate the synthesis and secretion of mucus by increasing blood flow in the membrane and contribute to the proliferation of the epithelium [37]. The above actions lead to bleeding and ulceration in the digestive system. It results in unpleasant ailments for patients like nausea, vomiting, stomach pain, and diarrhea [38]. IND is responsible for nausea and dyspepsia in 3–9% of patients, abdominal pain, diarrhea or constipations in 1–3% of patients, whereas other GI events are possible in less than 1% of patients [34].

In the kidney, NSAIDs inhibit the production of prostaglandins such as PGI2, PGE2, and PGD2. All of them play important roles in maintaining the normal kidney function [39]. PGE2 is a vasodilator agent and a major factor involved in salt excretion and water through the kidneys [40]. PGI2 also acts as a vasodilator agent, while PGD2 controls renal hemodynamics but has no effect on fluid and electrolyte secretion [41]. Inhibition of these hormones causes narrowing of blood vessels, reduction in blood volume, and in consequence kidney impairment. The lack of measures to counteract this condition can lead to tubular necrosis and acute renal failure [42]. With long-term use of NSAIDs, subclinical renal disorders such as decreased creatinine filtration or loss of urine concentrating capacity appear [43]. In most cases, these disorders resolve after NSAID withdrawal, however, there are reports suggesting persistence of these dysfunctions in some patients [44].

The negative effects of NSAIDs are also observed in the circulatory system, lungs, liver and skin and are the effect of inhibition of prostaglandins production, which disrupt the proper functioning of these systems/organs [45]. Therefore, in the late 1990s, scientists focused their attention on the development of COX-2 selective drugs, based on the observations that selective NSAIDs are less likely to cause negative side effects [46]. At the time of the development of selective NSAIDs, this hypothesis has not been confirmed yet by complete studies. In 2000, two large randomized controlled research projects have been conducted: CLASS (Celecoxib Long-term Arthritis Safety Study) and VIGOR (Vioxx Gastrointestinal Outcomes Research) and led to compare the safety of selective and non-selective NSAIDs [47,48]. The preliminary results of these studies have indicated a greater safety of using COX-2 selective NSAIDs, but after the publication of complete data by the Food and Drug Administration (FDA), it turned out that after taking them, the risk of cardiovascular complications (myocardial infarction, stroke ischemic, angina) is much higher [49,50]. Therefore, classic non-steroidal drugs are being used again, which, such as indomethacin, when administered in an appropriate dose still exhibit greater therapeutic benefits in comparison to the possible toxic side effects [51].

## 4. Novel Strategies for Indomethacin Delivery

### 4.1. Associations of Phospholipids with Indomethacin and Lipid Emulsions of Indomethacin

NSAIDs significantly alter the biochemical properties of lipid membranes such as bending stiffness and pore formations. IND attenuates the barrier properties of the lipid monolayer of the gastric mucosa by increasing its wettability. The effect of IND on the phase behavior of a mixed model bilayer mimicking biological membranes were deeply examined [52]. It has been reported that this drug significantly enhances immiscibility of saturated and unsaturated lipids and induces the formation of gel-phase domains in the mixed model membranes. Moreover, Fearon and Stokes discovered that adsorption of indomethacin to gel-phase phospholipids is endothermic and entropically driven, whereas adsorption to fluid-phase phospholipids is exothermic and enthalpically driven [53].

In the middle of the 1990s it has been demonstrated that NSAIDs could be associated with phosphatidylcholine and in this form do not induce gastric injury. It was reported that the toxicity of NSAID administrated as complexes with DPPC or related phospholipids was remarkably decreased whereas their antipyretic and anti-inflammatory activities enhanced [54]. In the research group of Lichtenberger, it has been proven that the pre-association of indomethacin with PC in the drug formulation prevents indomethacin from associating with mucus PC and disruption of its surface [55].

Moreover, it has been reported that indomethacin-PC associations prevented the metastatic spread of cancer cells in a syngeneic mouse model. Moreover, it has been evaluated that IND significantly inhibits the growth of the colon cancer cell line MC-26 at a concentration of 20 μmol/L whereas its complexes with PC are active at an even lower concentration of 8 μmol/L [56]. The mechanism of anticancer action of complexes of IND with PC was evaluated and it was discovered that IND promotes apoptosis [57]. The associations of indomethacin with PC showed also to be more potent inhibitor of PGE_2_ in comparison to the free form of IND [56].

High gastrotoxicity of indomethacin and side effects like headache and dizziness which this drug causes in the central nervous system (CSN) in most of patients limits its high therapeutic potential. Therefore, the possibilities of topical administration of indomethacin have been also studied. However, this way of administration has shown to be effective for some NSAIDs [58], but not for indomethacin, whose effectiveness was then at a lower level [58] probably due to its unfavorable physicochemical properties [59,60], which limits its ability to penetrate deep into the skin.

In order to overcome the mentioned limitations, the development of novel indomethacin topical formulations with higher skin penetration efficiency is required. Sakdiset et al. [61] as the solution for this problem proposed the ethosomes containing indomethacin. They developed the method of obtaining the ethosomes with good colloidal appearance using various concentrations of soybean phosphatidylcholine, ethanol, and dispersing agents and test them in an in vitro model on pig skin. Prepared ethosomes led to significantly higher permeation of IND through pig skin over 24 h than the commercial solution and the ethanolic solution of indomethacin. Moreover, ethosomes were stable in terms of physical appearance, drug content, and entrapment efficiency (EE) during storage at ambient temperature for 3 months, ensuring greater permeability of indomethacin through the skin than a commercial solution of this drug or a prepared ethanolic solution, which indicates that it is possible to use this form of formulation as transdermal carriers for this drug [61].

There are also reports about the usage of lipid emulsions as an effective system of delivering indomethacin to the body. The most interesting solution in this regard seems to be those where the drug is applied in self-emulsifying drug delivery system (SEDDS). The principle of this method is based on the phenomenon in which a mixture of lipids, surfactants, and cosolvents is emulsified in an aqueous environment under gentle mixing conditions that occur in the digestive process in the gastrointestinal tract [62]. The active ingredient is usually more soluble in such a mixture than in the oil itself. In the study testing the effectiveness of the SEDDS system in delivering a poorly water-soluble indomethacin, a sample containing 30% Tween 85 and 70% ethyl oleate was selected as an optimized formulation with high drug concentration, low surfactant concentration, and small particle size. Studies in an in vivo rat model showed a 57% increase in drug concentration in the body after oral administration in the SEDDS form compared to the free form of indomethacin. After a rectal administration of gelatin hollow type suppositories filled with the SEDDS, the drug concentration in the blood was maintained at 41% higher than in the case of indomethacin [63].

### 4.2. Liposomal Formulations of Indomethacin and Their Efficacy

In the recent years, the administration of the liposomal forms of indomethacin was tested as the formulations that increase the stability of the drug, enhance its therapeutic effects, extend the circulation time of the active substance in the body and also reduce its toxicity (Table 2) [64]. Due to the similar structure to biological membranes, the use of liposomes is not associated with the risk of antigenicity [65]. One of the first examples of indomethacin-containing liposomes come from 1988 [66]. It was proved that the encapsulation of indomethacin in the structure of the liposome before oral administration to rats significantly decreased gastrointestinal ulceration and in some cases, it completely eliminated it [66]. Soehngen and coworkers reported that observed lower gastrotoxicity could be the result of reduction in bile concentrations and formation of mixed micelles with bile acids. They have also pointed out that may be encapsulated drug inhibits the reduction in prostaglandin synthesis as a result of an alteration in drug presentation in the GI mucosa. An alteration in the reactivity of the drug with the external environment have been also observed by D’Silva and Notari [67] for liposomal formulation of indomethacin. These findings imply that a specific orientation for this molecule within a liposomal membrane may affect its reactivity with cellular components. These interactions of IND with lipids of liposomal bilayer were studied also by Hernández et al. [68], who reported that IND can be extensively encapsulated by association to bilayers if liposomes are positively charged by stearylamine (STE) presence.

In the next years the effect of lipid composition and size on the targeting potential of liposomes encapsulated IND in arthritic rats as well as methods for increasing their circulation time have been extensively studied by Srinath and coworkers [69,70,71]. Multilamellar vesicles (MLVs) were found to exhibit the highest drug release which is slow for positively charged stearylamine-containing liposomes. This effect of charge has been attributed to electrostatic interaction (hydrogen bonding) between the acid moiety of drug and the amine moiety of lipid. For liposomal formulations of IND the significantly higher anti-inflammatory activity was proved in comparison to the free form of this drug in both carrageenan-induced rat paw edema and adjuvant arthritis models [71]. It has been reported that the effectiveness of indomethacin-containing liposomes can be increased by adding polyethylene glycol to the surface of the phospholipid bilayer. This modification increases the half-life of the liposomes, thereby increasing the drug circulation time in the body, increasing the efficiency of its delivery four times [71].

The enhancement of absorption of IND by increasing the retention time at the absorption site and reducing the distance to the blood was reported for liposomes modified with chitosan [72]. In fasted rats, the absolute bioavailability of IND was 92.9% and 93.1% for the uncoated and chitosan-coated liposomes, respectively. For comparison the free drug solution and suspension showed the oral bioavailability only on the level of 50.5% and 37.1%, respectively. Moreover, the chitosan-coated liposomes were characterized by higher bioavailability (75.2%) even when were administrated with meals. According to the retention profiles in the GI tract segments the chitosan-coated liposomes remained longer in all the segments including stomach, duodenum, jejunum, and ileum than the uncoated ones indicating the contribution of mucoadhesive property to the enhanced bioavailability.

The safety of liposomal formulation of IND was evaluated in a pregnant mouse model [73,74]. Refuerzo et al. reported that encapsulation of IND in liposomes with oxytocin receptor antagonist (LIP-IND-ORA) reduces placental passage of the drug to the fetus [73] and specifically designed nanocarriers are capable to increase the function of the drug available to its intended site of action decreasing the fetal exposure to the drug. Surface modification of conventional liposomes was also reported by Nazeer et al. as the effective method for design formulations of indomethacin with better in vivo pharmacokinetic [75].

## 5. Bioconjugates of Indomethacin

Bioconjugation of therapeutics is an effective method for targeted drug delivery and reduction of their undesired side effects. Linkage of drug with other biomolecule via covalent bond offers many clinical benefits like enhanced disease-specific targeting, reduced toxicity, optimized pharmacokinetics, and improved efficacy, safety, and tolerability. Especially bioconjugates are attractive which include lipids as the building blocks. As hydrophobic compounds, lipids after passing through the intestinal walls, penetrate into the lymphatic system through which they connect with the circulatory system at the level of the thoracic vein. This way of transporting lipids in the circulatory system distinguishes them from hydrophilic compounds, which, after penetration into the intestinal walls, reach the blood vessels, and then are transported through the portal vein to the liver, and then to the general circulation. Giving a lipid character to drugs after their conjugation with lipid molecule increase their concentration in the body, prolong release and blood circulation [76]. Active molecules are released from bioconjugates only as a result of the action of endogenous enzymes present in the target tissues [71]. Clinical trials have shown that therapeutic compounds administered in the form of lipid drug conjugate (LDC) are characterized then by greater oral bioavailability and lower toxicity in relation to the free forms of drugs. Moreover, drug release from such combinations can be controlled, which limits the occurrence of potential side effects [77]. In recent years, the Food and Drug Administration (FDA) and the European Medicines Agency approved the first drugs produced in the form of lipid conjugates, which have been used especially in the treatment of diabetes, schizophrenia, and depression [78].

In the recent years, many attempts have been made to develop bioconjugates of indomethacin. Most of them are based on the lipids and would yield adequate anti-inflammatory potency after oral administration without adversely affecting gastric mucosa. Paris et al. [79] proposed the strategy of production of prodrugs of IND with usage of glycerides. The inspiration for the development of this type of conjugates were the promising results of biological activity of conjugates containing aspirin in the *sn*-2 position of triacylglycerol (TAG) skeleton, which ensured the appropriate level of salicylates in the blood excluding the occurrence of gastric irritation [80,81,82]. The series of triacylglycerides containing IND **1**–**5** and **6a**–**g** was synthesized and evaluated for anti-inflammatory activity in the rat paw carrageenin edema assay (Figure 3) [79]. Monoglyceride **1** turned out to be the same active as free indomethacin whereas 2-indomethacin monoglyceride **2** and 1,2-diindomethacin glyceride **3** exhibited to be four to six times less potent, respectively. Derivative **4** and **5** were described as inactive.

The most active derivative in the group of studied compounds was the 1,3-dialkanoyl-2-indomethacin glyceride **6a**–**g** (Figure 4). All of them expressed less gastrotoxicity and compounds **6a** and **6e** produced activity comparable to indomethacin when administrated on a daily basis to adjuvant arthritic rats. The calculated “therapeutic indexes,” expressed as the ratio between the ulcerogenic dose (UD_50_) and the effective dose (ED_50_), were higher for modified by IND glycerides than for free form of IND. The acute gastric irritating properties of **6a** and **6e** were seven to eight times lower than IND, resulting in 2.5- to 3-fold improvement in the ratio of antiedema activity to ulcerogenicity [81]. The next step is the development of new bioconjugation of IND Paris and Cimon, and also the synthesis of glyceride with indomethacin in the *sn*-1 position of glycerol skeleton in which two alcohol groups were esterified by decanoyl residues, but this derivative exhibited six times lower activity than the corresponding analogue with IND in the *sn*-2 position [82].

A novel approach to overcome the problem of ulcerogenicity of IND has been proposed by Ueda et al. [83]. In their study ester **7** an anti-inflammatory molecule was administered as part of a histamine H_2_-receptor antagonist, which works by stopping the histamine-dependent mechanism to increase the production of gastric acid (Figure 5). The aim of the authors was to develop a new indomethacin derivative that is able to inhibit the synthesis of prostaglandins and does not produce lesions in the stomach. The strategy adopted by the authors was based on the assumption that after the application of indomethacin in the form of an ester, its hydrolysis to the drug and the free form of the compound that is an H_2_-receptor antagonist in the digestive system, will result in the inhibition of the adverse side effects of the drug. The efficacy of the new prodrug was comparable with indomethacin. Ester **7** almost completely inhibits carragenin-induced hind-paw edema in rat at a very high dose of 230 mg/kg. The same effect was obtained after the dose of 100 mg/kg of IND, but in opposite to ester in this case gastric lesions were also observed. Moreover, the acute gastric lesioning properties of prodrug were 100 times less than that observed after administration of indomethacin, resulting in over a 20-fold improvement in the ratio of antiedema activity to ulcerogenicity [83].

Dvir and coworkers focused their efforts on the development of a new type of hybrid by covalent bonding the IND with phospholipids [84]. They synthesized a bioconjugate of phosphatidylcholine with IND (DP-155) as a new approach to overcome the problem of indomethacin toxicity (Figure 6). Obtained hybrid DP-155 was a mixture of phosphatidylcholines **8a** and **8b** containing palmitic or stearic acid in the *sn*-1 position, and an indomethacin molecule attached to the glycerol backbone via five-carbon linker in the *sn*-2 position [84]. The activity and the level of toxicity of the mixture of conjugates were tested in an in vivo model on male Sprague-Dawley rats which were supplemented daily with the novel prodrug and free form of indomethacin at equal dose range of 0.007 to 0.28 mmol/kg. The results showed that ulceration of the gastrointestinal tract with the administration of DP-155 occurred 10 times less often than with the administration of indomethacin and arose significantly later. Positive results were also obtained in studies of the level of DP-155 toxicity in the kidneys. Side effects of indomethacin, such as decreased urine output or an increased ratio of *N*-acetylglycosaminidase to creatinine ratio in it, were five-fold lower for DP-155 than for the free form of the drug [84].

In next step the activity of D155 was studied in the in vivo model on Tg2576 transgenic mice, an animal model used in research on AD. The effects of DP-155 conjugates and indomethacin were compared when they were administered to animals every 4 h for 3 days in two equimolar doses: 0.14 mmol/kg and 0.046 mmol/kg. The efficacy of these substances was measured by the loss of β-amyloid 40 and 42 deposits in the brain, which are proteins characteristic for Alzheimer’s disease because for people with AD the ability to remove β-amyloid from brain tissue is impaired. One of the more probable hypotheses explaining the causes of AD–the amyloid hypothesis–is that the disease is caused by amyloid deposits in the brain that interfere with impulse conduction and lead to neuronal death [85]. On the basis of the obtained results, it was noted that both substances did not affect the level of Aβ40 proteins and to a similar extent reduced the level of Aβ42 proteins. At the higher dose, both indomethacin and DP-155 turned out to be toxic, while at the lower dose only indomethacin showed such effect. The most surprising results were obtained after analysis of the pharmacokinetics of both substances. The half-life of indomethacin orally administrated in the form of prodrug DP-155 was determined to be 22 h in the blood serum and 93 h in the brain, respectively, while for free form of indomethacin it was 10 and 24 h. Additionally, it was observed that the ratio of indomethacin concentration in the brain to serum concentration was 3.5 times higher when the conjugate was administered. This indicates that despite the higher concentration of indomethacin in plasma after its administration as a drug, DP-155 gives a much higher relative concentration in the brain. This may mean that above a certain concentration of indomethacin in the plasma, its penetration into the brain is limited or even blocked [84]. This phenomenon is probably due to the known mechanism that limits the uptake of substances into the brain, which is a result of vasoconstriction and limited blood flow after administration of high doses of indomethacin [86,87]. The relatively constant concentration of indomethacin in the serum after the administration of DP-155 allows for less frequent dosing of the active substance, which protects the gastrointestinal tract and kidneys from excessive damage.

In the in vivo studies the effects of oral indomethacin, indomethacin in the form of prodrug and intravenous indomethacin infusion on the brain’s uptake of these substances were also compared. For this purpose, Dahan et al. [88] administered indomethacin to a group of rats orally in the form of the DP-155 prodrug and in the free form, and intravenously in the form of commercially available injections, measuring the level of the drug in the blood and brain of the animals. The reported concentrations of indomethacin in the brain were comparable with orally administrated indomethacin (0.45 μ/g), oral prodrug (0.3 μ/g), and intravenous indomethacin (0.31 μ/g). The corresponding plasma concentrations were 14.1, 4.1, and 4 μg/mL, respectively, after oral administration of free IND, prodrug indomethacin, and intravenous indomethacin infusion. However, similar to previous studies, the brain to serum indomethacin level ratio was 2.5 times higher after administration of the prodrug injection compared to the oral form. Thus, it was found that the distribution of indomethacin to the brain is directly dependent on the method of its administration, and that long-term systematic controlled drug release may provide the desired pharmacodynamic effects [88].

The authors investigated the mechanism of oral controlled release of IND from phospholipid prodrug and analyzed the influence of the linker length connecting the indomethacin molecule with the glycerol backbone of phosphatidylcholine on the level of degradation by the enzyme phospholipase A_2_ (PLA_2_), which catalyzes the ester bond hydrolysis reaction selectively at the *sn*-2 position of the phospholipids. Phosphatidylcholine derivatives containing a drug molecule attached at the *sn*-2 position to the glycerol backbone through 2-, 3-, 4- and 5-carbon chains were synthesized. The highest degree of hydrolysis was obtained for the derivative containing the 5-carbon linker. In this case, approximately 60% of the administered dose of conjugate was hydrolyzed to the free form of the drug. As the linker length decreased, the level of ester bond hydrolysis in the remaining conjugates decreased. The shorter linkers caused a 20-fold decrease in the subsequent indomethacin absorption. This can be explained by the steric effect that occurs in the case of short linkers, which probably does not occur with the 5-carbon linker. The phospholipid indomethacin conjugate (DP-155) was found as a potential novel mechanism for oral controlled release for this drug molecule and the studies showed that it is possible to control the release kinetics of drug from a prodrug by PLA_2_ [89].

It has also been confirmed that the DP-155 molecule can act in the brain tissue, contributing to the reduction of neurodegeneration related with the accumulation of beta amyloid (Aβ) peptides in cells and inflammation. After oral administration the cleavage in the GI tract resulted in constant low serum levels and low maximal concentration, facilitating 5–10-fold better GI and renal safety. Moreover, delivery of conjugate DP-155 did not cause local gastric toxicity, probably because it selectively inhibits COX-2 but not COX-1. Although the blood level of indomethacin after its initial administration as the DP-155 conjugate is lower than after an equimolar dose of the free form, the concentration of this drug in the brain remains at a similar level. Due to the unique PK, DP-155 demonstrates more than five-fold reduction in toxicity, but maintains effects in the central nervous system similar to that of IND. The in vitro and in vivo studies conducted so far by Dvir and coworkers have shown that the DP-155 molecule is a safer form of the drug and a good alternative for the treatment of Alzheimer’s disease and for use as an antipyretic and/or analgesic agent [90].

The design of the structure of therapeutic molecules being the substrates for endogenous PLA_2_ enzyme has become even more legitimate since the elevated level of one of the subtype of PLA_2_–sPLA_2_ was detected in the surrounding tumor microenvironment in human colorectal adenocarcinomas and in neoplastic prostate tissue [91,92]. Rosseto and Hajdu have developed a new synthetic phospholipid analogue as the substrate for sPLA_2_ which contained selected anti-inflammatory drugs. One of the presented prodrugs was oligoethyleneglycol-substituted in the *sn*-2 position by indomethacin (**9**) (Figure 7) [93].

## 6. Conclusions

This work presents an overview of the literature data about indomethacin developed during the past few decades, methods of reduction of its gastrotoxicity, and targeted delivery by formation of lipid formulations and bioconjugates. The study summarized in this review established lipid bioconjugates of indomethacin as promising candidate for future clinical studies and proved that lipids protect against acute intestinal epithelial injury caused by indomethacin and effectively increased the therapeutic efficiency of this drug.

Application of phospholipids and nanotechnology is a promising approach for further development of innovative formulation for indomethacin and other drugs from the group of NSAIDs and is expected to grow in the near future.

## Figures and Tables

**Figure 1 molecules-26-01576-f001:**
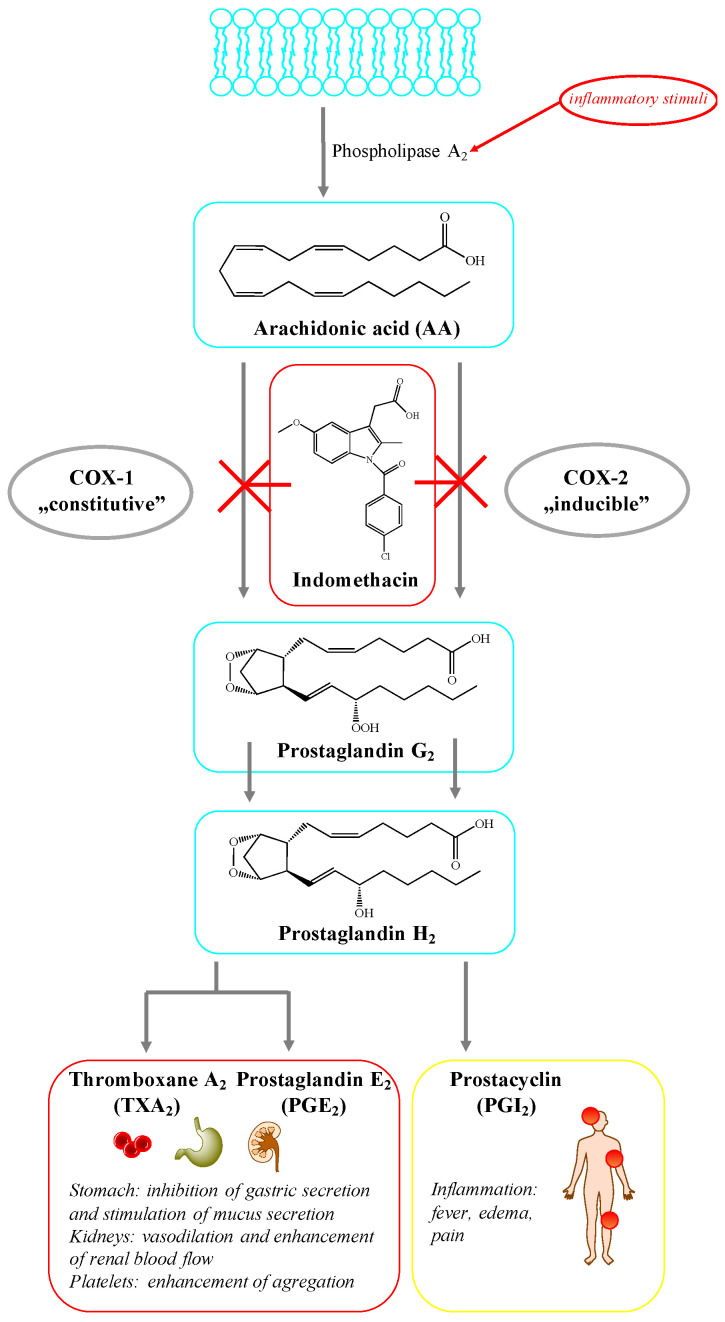
The mechanism of action of indomethacin.

**Figure 2 molecules-26-01576-f002:**
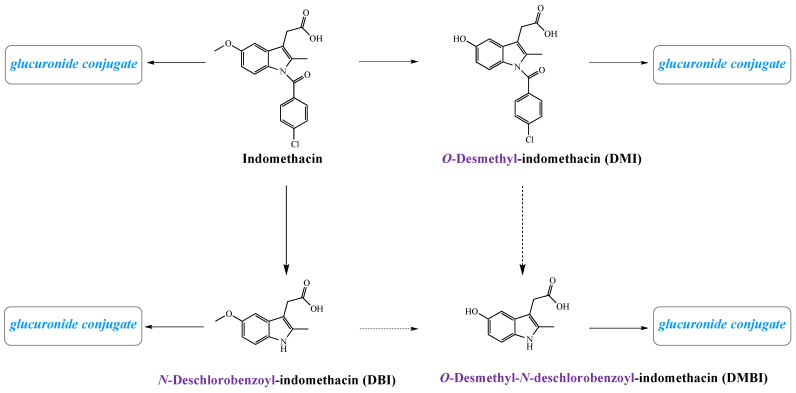
Metabolism of indomethacin in human.

**Figure 3 molecules-26-01576-f003:**
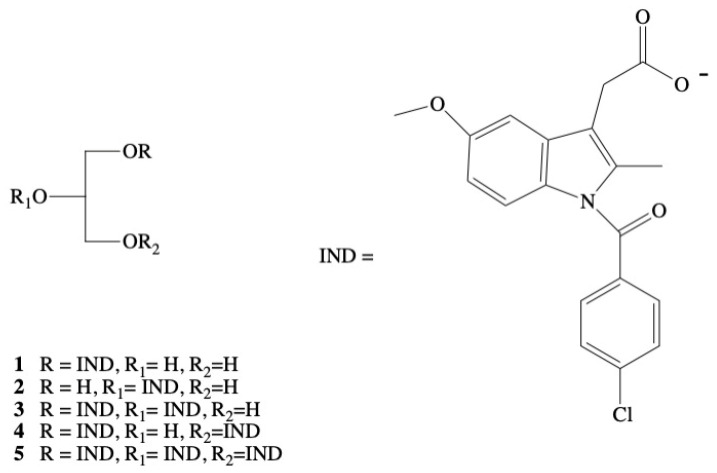
Chemical structure of triacylglycerides containing indomethacin **1**–**5**.

**Figure 4 molecules-26-01576-f004:**
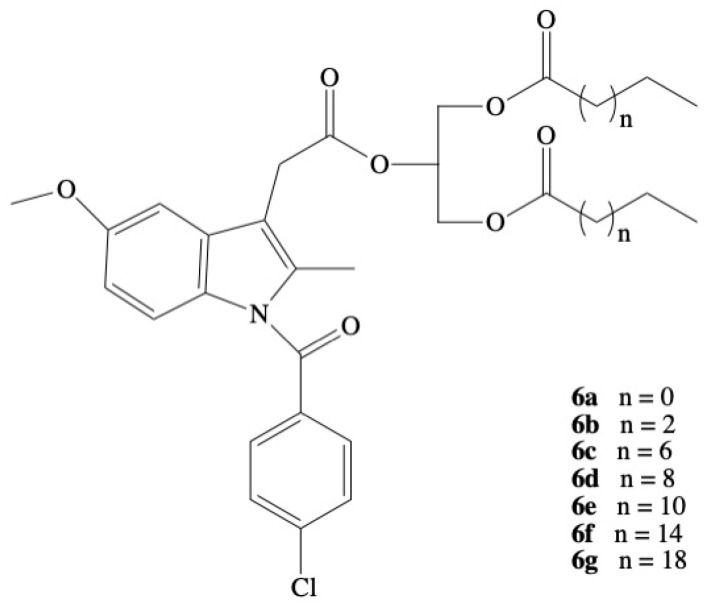
Chemical structure of 1,3-dialkanoyl-2-[1(*p*-chlorobenzoyl)-5-methoxy-2-methylindole-3-acetyl]glycerides **6a**–**6g**.

**Figure 5 molecules-26-01576-f005:**
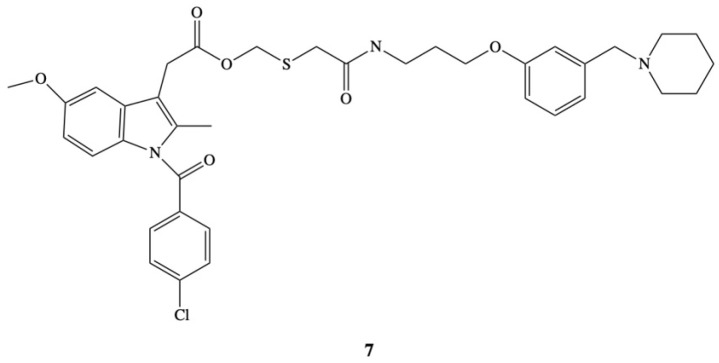
Chemical structure of 2-[*N*-[3-{3-(Piperidinomethyl)phenoxy}propyl]carbamoylmethylthiol]-ethyl 1-(*p*-chlorobenzoyl)-5-methoxy-2-methylindole-3-acetate (**7**).

**Figure 6 molecules-26-01576-f006:**
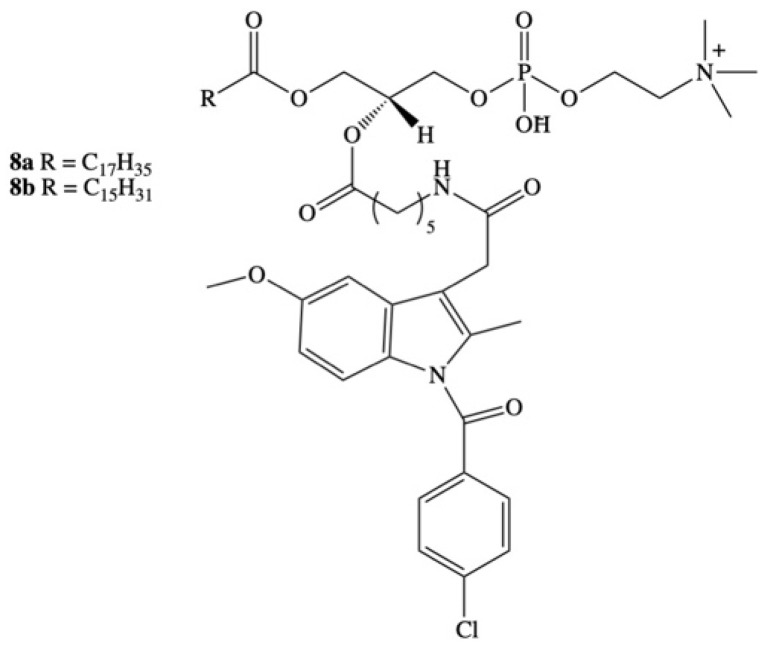
Chemical structure of phospholipid derivatives of indomethacin DP-155.

**Figure 7 molecules-26-01576-f007:**
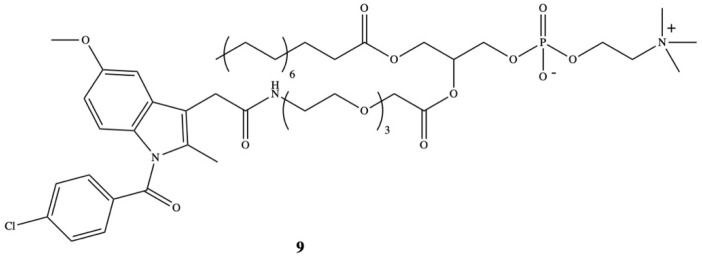
Chemical structure of 1-palmitoyl-2-(11′-N-indomethacincarboxylamino-3,6,9-trioxaundecanoyl)-*sn*-glycerophosphocholine (**9**).

**Table 1 molecules-26-01576-t001:** Anticancer activity of indomethacin and its mechanism of action in the in vitro and in vivo models.

Research Model	Active Dose	In Vitro/In Vivo Outcomes	Reference
Patients with desmoid tumors	100 mg/day in combination with ascorbic acid	Inhibition of a desmoid tumor/ postulate that lowering of cAMP inhibits cell growth	[18]
Patients with desmoid tumors	Alone or in combination with 5-fluorouracil and cyclophosphamide	Inhibition of suppressor T-cells	[19]
Dimethylhydrazine (DMH)-induced rats	2 mg/kg per day	Inhibition of formation of aberrant crypt foci (ACF)/significant inhibition of growth and development of tumors	[20]
Dimethylhydrazine (DMH)-induced rats	2 mg/kg per day	Reduction of 42% of number of ACF	[21]
Dimethylhydrazine (DMH)-induced rats (colorectal tumors)	2 mg/kg per day	Reduction of tumor number (83.5%), reduction of tumor volume (95%), increase of rate of apoptosis and reduction of proliferation in the S phase	[23]
Patients with head and neck cancer (stage III and IV)	75–100 mg/day	Tumor regression, increase of survival	[24]
Lung carcinoma cells mouse	10–20 μM 2 mg/kg per day	Reduction of cells at the S and G_2_/M phases and increase of cells at G_1_ phase inhibition of COX activity and effective in delaying the growth of both the primary tumor inoculate and of lung metastatic nodules	[25]
Human colon carcinoma (HT29, HCT116, Caco-2), lung adenocarcinoma (A549), cervical adenocarcinoma (HeLa) cells	400–1000 μM	Selective activation of dsRNA (double-stranded RNA)-dependent protein kinase PKR in a cyclooxygenase-independent manner, rapid phosphorylation of eIF2α and inhibition protein synthesis in carcinoma, induction of apoptosis	[26]
Human epidermoid carcinoma (A431)	1–10 μM	Inhibition of cancer cell migration by influencing calcium mobilization and focal complex formation	[27]
Melanoma cells (A375)	1–300 μM	Promotion of TRIAL-induced cell death and apoptosis, induction of cell surface expression of death receptor 5 (DR5)	[28]
Prostate cancer cells	2.5–10 μM	Inhibition of activity of enzyme in the steroidogenesis pathway AKR1C3 through binding with its active site, strong selectivity for AKR1C3 at 8.2 μM over AKR1C1 and AKR1C2 (over 100 μM), inhibition of the levels of intracrine androgens in C4-2B MDVR cells and CWR22Rv1 cells and suppression of prostate cancer tumor growth	[29]

**Table 2 molecules-26-01576-t002:** Liposomes formulations of indomethacin.

Liposomal Type	Physiochemical Characteristics	Study Model	In Vivo Outcomes	Reference
IND encapsulated into egg PC (EPC) monophasic vesicles (MPV) and into stable plurilamellar vesicles (SPLV)	Spherical structures, size range of 0.5 μm	Male Wistar rats	EPCMPV containing IND (4 mg/kg) reduced gastric and intestinal ulceration, anti-inflammatory effect with a dosage ranging between 0.5 and 4 mg/kg	[66]
IND encapsulated in liposomes prepared with usage of various phospholipids (PC, PE, PG), stearylamine (SA) and cholesterol (CH)	The highest encapsulation efficiency 32% for liposome composition PC:CH:SA (1:0.5:0.1 molar ratio),	Male Wistar rats	C_max_ in the liver delayed from 1 h for free drug to 4 h for encapsulated form, localization in the liver was greatest for liposomes PC:CH:PG (1:0.5:0.2 molar ratio), this composition is the optimum for targeting arthritic joints	[69]
Long-circulating liposomes (S-LI)	Encapsulation efficiency 52–55%, liposome composition PC:CH:PE-PEG (1:0.5:0.16)	Male Wistar rats	Better pharmacokinetic profile (AUC_0−t_ 1454.62 ± 92.85 μg/mL/h, elimination half-life 25.42 ± 4.32 h and clearance 0.82 ± 0.15 mL/h, MRT 36.36 6.25 h) than free IND (AUC_0−t_ 490.95 ± 31.28 μg/mL/h, elimination half-life 10.28 ± 0.25 h and clearance 4.20 0.33 mL/h, MRT 13.27 0.49 h) Increased anti-inflammatory activity, less ulcer index	[70]
Chitosan-coated liposomes	Liposome composition DSPC:DCP:CH (8:2:1) coating with chitosan, Liposomal dispersion, Size 270–310 nm	Rats	Prolonged intestinal transit, delayed drug release profile	[72]
Multilamellar liposomes	Size 159.8 nm Polydispersity index < 0.069 Encapsulation efficiency 93%	Rats	Reduction of the drug levels within the fetus by 7.6-fold yet maintained its pharmacologic effects	[73]
IND loaded in liposomes with oxytocin receptor antagonist (LIP-IND-ORA)	Size 154.2 nm Zeta potential −21.2 mV Encapsulation efficiency 93%	Rats	Uterine to fetus IND concentration ratio was 4-fold higher for liposomes than for free drug, encapsulation of IND does not alter the pharmacological activity of drug	[74]
Sterically stabilized liposomes	Zeta potential −35.3 mV Encapsulation efficiency 64.04–79.54%	Rats	Increased in vitro drug release in comparison to the conventional liposomal formulation, better in vivo circulation time and enhanced mean percentage edema decrease for stealth liposomes in comparison with conventional liposomes and drug, higher stability (3 months)	[75]

## Data Availability

Data is contained within the article.

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
