# Peer review of "Lipid Formulations and Bioconjugation Strategies for Indomethacin Therapeutic Advances"

_molecules, 2021, doi:10.3390/molecules26061576_

Round 1

Reviewer 1 Report

The review describes very well the classical formulation methods for indomethacin.

Nevertheless, I have a few comments to address to the authors:

  1. Few typing errors should be corrected:

Figure 1 : “vasoldilasion”

Line 145: In addition, “an interesting are” studies indicating that

Line 369:  while for free form of indomethacin ”it” 10 and 24 hours

Line 396: The “Authors” investigated

  1. The discussed examples in this review are not so recent.

 In the last paragraph of the review you discuss about nanotechnology as promising approach. You should add at least one section with the last developments for indomethacin delivery using different innovating carriers.

Author Response

4 March, 2021

Prof. Dr. Andrea Trabocchi

Dr. Elena Lenci

Guest Editors of Molecules

Dear Editor in Chief of Molecules,

We would like to thank you for the opportunity to revise our paper on “Lipid formulations and bioconjugation strategies for indomethacin therapeutic advances” (molecules-1126754). The offered suggestions have been immensely helpful for us. We have included the Reviewer comments and responded to them individually, indicating exactly how we addressed each concern or problem and describing the changes we have made. All changes were highlighted in yellow in revised version of the manuscript.

According to the Reviewer 1 comments:

Few typing errors should be corrected:

Figure 1 : “vasoldilasion”

Line 145: In addition, “an interesting are” studies indicating that

Line 369:  while for free form of indomethacin ”it” 10 and 24 hours

Line 396: The “Authors” investigated

Response: We would like to thank you Reviewer for indicated errors, all of them were corrected and whole manuscript was checked and corrected paying more attention on grammar and language errors.

The discussed examples in this review are not so recent. In the last paragraph of the review you discuss about nanotechnology as promising approach. You should add at least one section with the last developments for indomethacin delivery using different innovating carriers.

Response: We fully agree with the Reviewer therefore the paragraph 4 has been reorganized and we added section 4.2 about liposomal formulations of indomethacin developed as promising delivery system for this drug (page 8-9).

According to the Reviewer 2 comments:

In the second part about the therapeutic activity, it is better to organize a comprehensive figure to include the main described reviewed works. It is better to guide the readers. A similar strategy can be done for the other parts. In the third part, it is better to decompose them in more detailed small parts rather than include them in the same paragraph from line 163-line 202.

Response: We agree with the Reviewer that some sections needed decomposition. According to the Reviewer suggestion we reorganized the section 2 and added a comprehensive Table 1 which includes described literature data making them clearer for the Readers. In section 3 we used additional separation by dividing the text with new paragraphs. We did not decide here for more advance decomposition because the most side effects of indomethacin and other drugs from NSAIDs mainly refer to gastrointestinal tract and kidney which are now presented in separate paragraphs.

For the lipid formulation, it is better to describe the mechanism of the interactions of IND and the lipid membrane or liposome. Additionally, more descriptions are preferred about the releasing process of IND from the liposome/lipid formulations after its entry into the human body.

Response: We agree with the Reviewer that description of interactions of indomethacin with lipids and liposomes should be presented in the manuscript. We reorganized paragraph 4 and added new information about mentioned interactions as well as the liposomal formulations of indomethacin developed as promising delivery system for this drug (page 6-9).

In the Bioconjugates part, it is highly recommended to reorganize the figures and add more detailed figures about the reviewed papers.

Response: Reviewer has right therefore we added new figures for synthesized derivatives of indomethacin with triacylglycerol and detailed information (section 5 page 10-11).

The paper is also required proofreading about the language, here I just point out a few of them as an example: 

Line 14 and line 19: application-> applications

Line 27: drug-> drugs

Line 314 Ueda et al -> Ueda et al [citation]

Response: We would like to thank you Reviewer for the indicated errors, all of them were corrected and whole manuscript was checked and corrected paying more attention on grammar and language errors.

We would like to express once again our thanks to Reviewers for very valuable comments, which help us to improve our manuscript. We hope that our explanations and corrections are sufficient, and will be accepted.

With kind regards,

Anna Gliszczyńska

Reviewer 2 Report

In this reviewer, the authors give a comprehensive review of the drug of IND, including its chemical structure, efficacy, side effects, and corresponding lipid formulation and bioconjugation strategies.  Overall, this paper is organized fine and will be interesting to the readers. While I have some suggestions as follows:

  • In the second part about the therapeutic activity, it is better to organize a comprehensive figure to include the main described reviewed works. It is better to guide the readers. A similar strategy can be done for the other parts.
  • In the third part, it is better to decompose them in  more detailed small parts rather than include them in the same paragraph from line 163-line 202.
  • For the lipid formulation, it is better to describe the mechanism of the interactions of IND and the lipid membrane or liposome. Additionally, more descriptions are preferred about the releasing process of IND from the liposome/lipid formulations after its entry into the human body.
  • In the Bioconjugates part, it is highly recommended to reorganize the figures and add more detailed figures about the reviewed papers.
  • The paper is also required proofreading about the language, here I just point out a few of them as an example:

         Line 14 and line 19: application-> applications

         Line 27: drug-> drugs

        Line 314 Ueda et al -> Ueda et al [citation]

Author Response

(The authors gave the same response as above.)

Round 2

Reviewer 1 Report

The authors answered to all my questions and comments.

The manuscript can be accepted in the revised form.

Reviewer 2 Report

The authors already addressed my questions.